# Correlation between Electrodiagnostic Study and Imaging Features in Patients with Suspected Carpal Tunnel Syndrome

**DOI:** 10.3390/jcm11102808

**Published:** 2022-05-16

**Authors:** Jae Min Song, Jungyun Kim, Dong-Jin Chae, Jong Bum Park, Yung Jin Lee, Cheol Mog Hwang, Jieun Shin, Mi Jin Hong

**Affiliations:** 1Department of Rehabilitation Medicine, Konyang University College of Medicine, Daejeon 35365, Korea; green4leaf@naver.com (J.M.S.); kirshna0615@hanmail.net (J.K.); coehdwls@hanmail.net (D.-J.C.); jbocean@hanmail.net (J.B.P.); eutravel@kyuh.ac.kr (Y.J.L.); 2Department of Radiology, Konyang University College of Medicine, Daejeon 35365, Korea; rad@kyuh.ac.kr; 3Department of Biomedical Informatics, Konyang University College of Medicine, Daejeon 35365, Korea; jeshin@konyang.ac.kr

**Keywords:** carpal tunnel syndrome, electrodiagnosis, X-rays, ultrasonography, median nerve

## Abstract

Electrodiagnostic studies (EDXs) are the confirmative diagnostic tool for carpal tunnel syndrome (CTS). Previous studies have evaluated the relationship between EDXs and ultrasonography (US) but not with X-rays. Recently, many studies on the diagnostic value of X-rays in various diseases have been reported, but data on CTS are lacking. We evaluated the relationship between electrodiagnostic parameters and roentgenographic and ultrasonographic features in CTS and investigated the usefulness of X-rays and US for CTS. This retrospective study included 97 wrists of 62 patients. All patients with suspected CTS underwent EDXs, wrist US, and wrist X-rays. The CTS patients were classified into mild, moderate, and severe groups. The roentgenographic features included the ulnar variance (UV) and the anteroposterior diameter of the wrist (APDW), and the ultrasonographic features included the flattening ratio (FR) and the thickest anteroposterior diameter of the median nerve (TAPDM). Most EDX parameters showed significant correlations with roentgenographic and US features. The electrodiagnostic severity was also correlated with all imaging features. Therefore, both wrist X-rays and wrist US can be useful for the diagnosis of CTS as supplements to EDXs.

## 1. Introduction

Carpal tunnel syndrome (CTS), or entrapment neuropathy of the median nerve at the wrist, is a common condition associated with numbness, tingling, pain that frequently worsens at night, and atrophy in the thenar region as the typical symptoms [1]. Thus, a patient with these symptoms is considered to have CTS, and women are much more apt to have this condition than men [2].

Electrodiagnostic studies (EDXs) are the confirmative diagnostic tool and are also used for severity grading, but they do not provide anatomic information at the wrist. Therefore, ultrasonography (US) has been used to visualize the median nerve and its surrounding anatomic structures [3,4]. Several studies have been conducted on various US features useful for the diagnosis of CTS, including the cross-sectional area (CSA), flattening ratio (FR), palmar bowing, thickest anteroposterior diameter of the median nerve (TAPDM), and the wrist-to-forearm ratio [4,5,6,7,8,9,10]. However, it is still unknown which one is the most indicative of CTS. Buchberger et al. [6] first reported that compared to normal wrists, the median nerves of CTS patients were significantly flattened. Duncan et al. [7] revealed significant differences in the FR of the median nerve and the TAPDM between CTS patients and controls.

There have been insufficient studies showing that simple X-rays have a diagnostic value for CTS, but they have the advantage of providing morphological images easily and inexpensively. Considerable progress in the development of radiology technology has been made, and studies on the diagnostic value of X-rays have recently been reported [11]. Among five roentgenographic features, Ikeda et al. [12] reported that there was a statistically significant difference in ulnar variance (UV) between CTS patients and controls. Therefore, additional research on the usefulness of plain radiography for the diagnosis of CTS is required.

A few other studies have evaluated the relationship between EDXs and US features. However, to our knowledge, no study has evaluated the relationship between EDXs and roentgenographic features. Therefore, we were interested in (1) examining the relationship between EDX parameters and roentgenographic and US features in patients with suspected CTS and (2) confirming the usefulness of roentgenographic and US features as a tool for diagnosing CTS.

## 2. Materials and Methods

### 2.1. Subjects

This study was designed as a retrospective chart review. We collected the data of 68 patients between January 2019 and May 2021. The patients who visited the outpatient department of rehabilitation medicine in a single center with some or all symptoms of CTS and underwent EDXs, wrist US, and wrist simple X-rays were selected for the study. The symptoms were sensory abnormalities in median nerve distribution, nocturnal pain, atrophy in the thenar region, and several positive provocative tests (Tinel sign, Phalen’s maneuver, and reverse Phalen’s maneuver) [13,14,15]. The exclusion criteria were as follows: (1) Patients with a history of wrist surgery or injections and any upper extremity trauma; (2) patients with neurologic diseases, such as diabetic polyneuropathy, brachial plexopathy, ulnar neuropathy, proximal median neuropathy (entrapment of the ligament of Struthers, pronator syndrome, anterior interosseous nerve syndrome), cervical radiculopathy, and rheumatic diseases; and (3) patients with hereditary or metabolic diseases that can cause peripheral neuropathy. Six patients were excluded due to insufficient wrist images. Finally, 97 wrists of 62 patients were enrolled in this study (Figure 1). The study was approved by the Institutional Review Board of Konyang University College of Medicine (IRB no. 2021-06-010).

### 2.2. Electrodiagnostic Studies

EDXs were conducted using Natus Synergy on a Nicolet EDX machine. All patients underwent needle electromyography and routine nerve conduction studies (NCSs) with an antidromic technique, including median and ulnar NCSs [16]. The temperature of both hands was measured and maintained between 32 °C to 34 °C. For the sensory NCS of the median nerve, a surface ground electrode was placed over the dorsum of the hand. A pair of surface recording electrodes were placed in line over the index finger at an inter-electrode distance of 4 cm. Standard stimulation was conducted at two sites that were 14 cm proximal to the active electrode (wrist) and 7 cm proximal to the active electrode (palm). For the motor NCS of the median nerve, the belly-tendon method was used for recording. A surface ground electrode was placed over the same site as that used in the sensory NCS. A surface active electrode was placed over the center of the belly of the abductor pollicis brevis (APB) muscle, and a surface reference electrode was placed distally over the tendon of the APB muscle. Standard supramaximal stimulation was conducted at the wrist. The sensory NCS parameters collected were the onset latency, peak latency, baseline-to-peak amplitude, and conduction velocity. The motor NCS parameter collected was the distal motor latency.

When the routine NCS results were normal, we performed three additional Preston’s median-versus-ulnar comparison studies [17]. In the palmar mixed comparison study, the median mixed nerve latency across the palm was compared to the adjacent ulnar mixed nerve latency using identical distances between the stimulation and recording sites. In the digit 4 comparison study, the median sensory latency recording of digit 4 was compared to the ulnar sensory latency recording of digit 4, using identical distances between the stimulation and recording sites. In the lumbrical-interossei comparison study, the median motor latency recording of the second lumbrical was compared to the ulnar motor latency recording of the interossei using identical distances between the stimulation and recording sites. A very mild CTS score (grade 1 on the Bland scale) was assigned when two or more of these three sensitive studies were positive.

The severity of the patients was classified according to the Bland scale using the EDX results [18]. Grade 0 (normal) indicated no neurophysiological abnormality in the sensory and motor conduction studies. Grade 1 (very mild CTS) indicated that abnormalities were detected in two or more sensitive tests. Grade 2 (mild CTS) indicated slowing sensory nerve conduction velocity and normal distal motor latency (<4.5 ms from the wrist to the APB muscle). Grade 3 (moderately severe CTS) indicated preserved sensory potential and slowing distal motor latency (>4.5 ms and <6.5 ms). Grade 4 (severe CTS) indicated absent sensory potential and slowing distal motor latency (>4.5 ms and <6.5 ms). Grade 5 (very severe CTS) indicated absent sensory potential and slowing distal motor latency (>6.5 ms), and grade 6 (extremely severe CTS) indicated decreased surface motor potential from the APB (<0.2 mV).

The above severity grades were also reclassified into four severity groups [19]. A severity grade of 0 was the control group, severity grades 1 and 2 were classified as the mild group; a severity grade of 3 was assigned to the moderate group; and severity grades of 4, 5, and 6 were classified as the severe group (Table 1).

### 2.3. Wrist X-rays

All patients underwent simple X-rays of the posteroanterior and lateral view of the wrist. To obtain the posteroanterior view, the elbow was flexed 90°, the forearm was pronated, and the wrist was in a neutral position. The UV was defined as the distance between horizontal lines (that were perpendicular to the long axis of the radius/ulna) drawn from the distal ulnar and radial articular surfaces (at the level of the distal radioulnar joint) on a posteroanterior view (Figure 2A) [12,20]. To obtain the lateral view, the elbow was flexed 90° and adducted against the trunk, and the wrist was in a neutral position. The anteroposterior diameter of the wrist (APDW) was defined as the distance between the volar and dorsal edge of the distal radius on a lateral view (Figure 2B) [12]. Two raters performed the measurements without other information on the patients, and the mean of the measurements was used in the analyses.

### 2.4. Ultrasonography

An experienced radiologist who was blinded to all of the patient’s results conducted the US evaluations using an ultrasound system with a 5–15 MHz linear transducer (GE LOGIQ E9; General Electrical Healthcare, China). No additional force was applied other than the weight of the probe. The FR of the median nerve was calculated as the ratio of the nerve’s major axis to its minor axis at the pisiform bone level on the transverse view (a/b) (Figure 2C) [7]. The TAPDM, including the hypoechogenic median nerve and hyperechogenic nerve sheath, was measured between the carpal tunnel inlet and outlet on the longitudinal view (Figure 2D) [9]. Two raters performed the measurements without other information on the patients, and the mean of the measurements was used in the analyses.

### 2.5. Statistical Analyses

Statistical analyses were performed using SPSS statistical software version 28.0 for Windows (IBM, Armonk, NY, USA). First, to examine differences in the distribution of demographic characteristics, we used the chi-square test for categorical variables and the analysis of variance (ANOVA) for continuous variables. Second, Pearson’s correlation coefficient was used to assess the relationship between EDXs and roentgenographic features, EDXs and US features, and roentgenographic and US features. Because sensory potentials were absent in the severe group and could not be quantified, the correlation analysis could not include the severe group, so the analysis included all except the severe group. Third, to examine differences in roentgenographic and ultrasonographic features in CTS patients and controls, we used *t*-test. Furthermore, to examine differences in roentgenographic and ultrasonographic features between the four severity groups, we used one-way ANOVA. If the result of ANOVA was statistically significant, Scheffe’s method was additionally used for multiple comparisons in post hoc analysis. Lastly, logistic regression analysis was conducted to evaluate the independent, related variables of CTS in the four imaging features and to determine the odds ratio (OR) and corresponding 95% confidence intervals (95% CI). Stepwise backward elimination was used to identify the most significant predictor of CTS. The diagnostic value of the imaging features was evaluated by the area under the receiver operator characteristics (ROC) curve. All analyses were tested at the significance level of 0.05.

## 3. Results

### 3.1. Demographic Characteristics

The demographic characteristics of the subjects are shown in Table 2. The subjects were 17 males and 45 females. The mean ages of the participants in the four groups were 41.13 ± 14.46, 53.50 ± 9.05, 59.80 ± 8.43, and 62.89 ± 8.43 years, respectively. Age and sex were significantly different between the four groups, and the ratio of left hands to right hands was not significantly different.

### 3.2. Relationship between Electrodiagnostic Parameters and Roentgenographic and Ultrasonographic Features

Pearson’s correlation coefficient is shown in Table 3 and Table 4. First, both the UV and APDW, which were roentgenographic features, showed statistically significant correlations with all EDX parameters, except for between UV and distal motor latency. Second, the FR and TAPDM showed statistically significant correlations with all EDX parameters. Especially, the correlation coefficients between sensory onset latency and the FR (r = 0.772), sensory peak latency and the FR (r = 0.772), sensory conduction velocity and the FR (r = −0.725), and distal motor latency and the FR (r = 0.703) were relatively high. Third, there were significant relationships between all roentgenographic features and all US features.

### 3.3. Differences in Roentgenographic and Ultrasonographic Features between CTS Patients and Controls

The imaging features in the CTS patients and controls are summarized in Table 5. There were significant differences in all four imaging features between the two groups. The mean UV was 1.45 ± 1.89 in the CTS patients and −0.01 ± 1.55 in the controls. The mean APDW was 23.81 ± 2.34 mm in the CTS patients and 22.14 ± 2.07 mm in the controls. The mean FR was 3.53 ± 0.52 in the CTS patients and 2.81 ± 0.28 in the controls. The mean TAPDM was 2.44 ± 0.46 mm in the CTS patients and 1.97 ± 0.35 mm in the controls.

### 3.4. Differences in Roentgenographic and Ultrasonographic Features between the Four Severity Groups

The roentgenographic and the US features in the four severity groups are shown in Table 6. The mean UV and standard deviation were −0.01 ± 1.55, 1.69 ± 2.18, 0.96 ± 1.61, and 1.41 ± 1.61 in the control, mild, moderate, and severe groups, respectively. The APDW was 22.14 ± 2.07 mm, 23.11 ± 2.29 mm, 23.85 ± 1.93 mm, and 25.09 ± 1.93 mm, respectively. The FR was 2.81 ± 0.28, 3.24 ± 0.41, 3.80 ± 0.48, and 3.87 ± 0.48, respectively. The TAPDM was 1.97 ± 0.35 mm, 2.34 ± 0.39 mm, 2.49 ± 0.52 mm, and 2.58 ± 0.52 mm, respectively. All imaging features showed statistically significant differences.

Scheffe’s multiple comparison test was conducted for post hoc analysis. There was a significant difference in UV between the control and severe groups. There was a significant difference in APDW between the control and moderate groups. In the FR, there were significant differences between all subgroups except for the moderate and severe groups. There were significant differences in the TAPDM between the control and all CTS groups. 

### 3.5. Factors Related to Carpal Tunnel Syndrome

The OR and 95% CI in univariate logistic regression of the four imaging features adjusted for baseline age values are presented in Table 7. Neither UV (*p* = 0.05, OR 1.43, 95% CI 1.00–2.04) nor APDW (*p* = 0.237, OR 1.16, 95% CI 0.91–1.48) were significantly related to CTS. However, the FR (*p* < 0.001, OR 86.52, 95% CI 9.26–808.83) and the TAPDM (*p* < 0.002, OR 15.33, 95% CI 2.78–84.62) were significantly associated with CTS. The ROC curves are shown in Figure 3. The area under the curve (AUC) of the UV and the APDW was 0.832 (*p* < 0.001) and 0.805 (*p* < 0.001), respectively. The AUC of the FR and the TAPDM was 0.927 (*p* < 0.001) and 0.889 (*p* < 0.001), respectively. Table 7 also shows the multiple logistic regression analysis results using backward elimination. The FR (*p* = 0.001, OR 52.52, 95% CI 5.50–501.73) and the TAPDM (*p* = 0.032, OR 8.91, 95% CI 1.20–65.97) were significant variables, so they remained in the model.

## 4. Discussion

EDXs have been widely used as the standard test for diagnosing CTS. Routine NCS or various comparison studies [21] can confirm electrophysiologic abnormalities of the median nerve within the carpal tunnel and the location of the lesion can be confirmed using the inching technique [22]. However, due to the disadvantage of not being able to obtain information on the anatomic structure of the wrist along with pain experienced during the examination, several imaging tests, including magnetic resonance imaging (MRI), computed tomography (CT), US, and X-rays, have been used together. Among them, both the US and X-rays have the common advantages of being easy, fast, non-invasive, and painless for the patient during the examination. Buchberger et al. [5,6] first reported the usefulness of US features such as the CSA, FR, and palmar bowing for CTS based on wrist MRIs, and Ikeda et al. [12] revealed the usefulness of roentgenographic features for CTS. However, they did not investigate the relationship between imaging features and various EDX parameters, which is the difference and uniqueness between this study and other previous studies. Our study demonstrated that all EDX parameters were correlated with all roentgenographic and US features, with just one exception between distal motor latency and the UV. In the post hoc analysis (Table 6), there was a significant difference in UV only between the control group and the severe group. Therefore, excluding the severe group from the correlation analysis could have caused this result. The sensory onset latency and sensory peak latency showed positive correlations with all imaging features, and the sensory amplitude and sensory conduction velocity showed negative correlations with all imaging features.

The UV represents the relative length of the ulna compared to the radius [23]. Cha et al. [24] reported a significant relationship between a decreased CSA around the distal radioulnar joint and a positive UV in CTS patients, supporting the importance of a positive UV in the development of CTS. Our UV findings were consistent with those of Ikeda et al. [12]. CTS patients showed a significantly high UV value compared to the controls. This finding suggests that although UV refers to the state of the extra space in the carpal tunnel and does not directly impact the median nerve, an imbalance in the distal radioulnar joint may be involved in the development of CTS [12,24].

The APDW is a popular and simple measurement of wrist size, estimating the carpal tunnel size. Carpal tunnel size has been a controversial risk factor for CTS. Bleecker et al. [25] and Dekel et al. [26] found that the carpal tunnel was smaller in CTS patients than in controls. Conversely, Winn et al. [27] reported that CTS patients had a larger carpal tunnel area than matched controls. Uchiyama et al. [28] showed that the proximal and distal carpal tunnel areas were significantly larger in mild-to-moderate and severe CTS patient groups than in the control group except for the extreme CTS group. The results of our study were similar. The APDW was larger in CTS patients than in the controls. Considering both the above findings, we suggest that there are other anthropometric risk factors or work-related factors, and further studies are needed.

Various studies have confirmed localized swelling and flattening of the median nerve in CTS patients by US or MRI [5,6,29,30]. Duncan et al. [7] reported an FR of 3.2 in CTS patients and 2.7 in controls by US at the level of the pisiform bone, which is usually the level of the maximum swelling of the median nerve. Additionally, they reported a TAPDM of 2.2 mm in CTS patients and 1.8 mm in controls by US. Uchiyama et al. [28] showed that the FR was significantly larger at the pisiform bone level in the mild-to-moderate, severe, and extreme CTS groups than in the control group based on MRI. Kim et al. [9] revealed that there were statistical differences in the TAPDM between CTS patients and controls. Likewise, we revealed that the FR and TAPDM values in CTS patients were larger than in the controls. Therefore, we also confirmed the swelling and flattening of the median nerve in CTS patients.

Identifying the factors related to CTS can be helpful in diagnosing CTS more precisely. The logistic regression analysis results showed that two US features were significantly related to CTS, and a higher FR was the most predictive factor for CTS. We speculate that this was due to the advantage of US, which can directly show the swelling and flattening of the median nerve. As reported in other studies, we also found that the US features could be useful for diagnosing CTS. Notably, the results of our study confirmed that the roentgenographic features were correlated with the EDXs and US features, and there was a significant difference between the CTS patient group and the control group. X-rays have limitations in that they mainly show bony structures and cannot show vascularity and tendons. However, they also have several advantages. They are relatively inexpensive compared to other imaging studies, and if they are conducted according to an accurate position and protocol, X-rays can be a more objective test than US, which can be relatively subjective depending upon the examiner.

This study had some limitations that should be considered. First, the retrospective nature was a major limitation. Due to the lack of data, we could not include information on CTS risk factors, such as the body mass index, wrist circumference, and participant occupation. This may have limited the generalizability of the results. Second, there were relatively fewer subjects, a higher percentage of female patients, and this was a single-center study. Particularly, the sample size of the control patients was small, and they were relatively young. Thus, these factors may have affected the logistic regression analysis, yielding a relatively high OR and wide 95% CI range. However, the fact that CTS occurs more frequently in women and between the ages of 50 and 60 should be taken into account. Third, although it is known that age affects nerve conduction velocity and waveform morphology, there were significant differences in age between the four groups. To compensate for this, we adjusted for baseline age values in the logistic regression analysis. Finally, no inter-rater reliability test was performed. However, two raters performed the measurements, and the average of the measured values was used. Despite these limitations, to our knowledge, this was the first study to report associations between EDXs and roentgenographic features. Further studies with a larger number of subjects are necessary to confirm and generalize our findings.

## 5. Conclusions

This study showed that most EDX parameters were correlated with roentgenographic and ultrasonographic features, and the electrodiagnostic severity was correlated with all wrist imaging features. Although the US features showed statistically better results than the roentgenographic features, depending upon the situation, X-rays can also be useful for diagnosing CTS. Therefore, we recommend considering both wrist X-rays and wrist US for patients with CTS symptoms to supplement EDXs. This will make the diagnosis of CTS more accurate. Furthermore, a larger follow-up study is necessary to reinforce the clinical effectiveness of wrist X-rays in CTS.

## Figures and Tables

**Figure 1 jcm-11-02808-f001:**
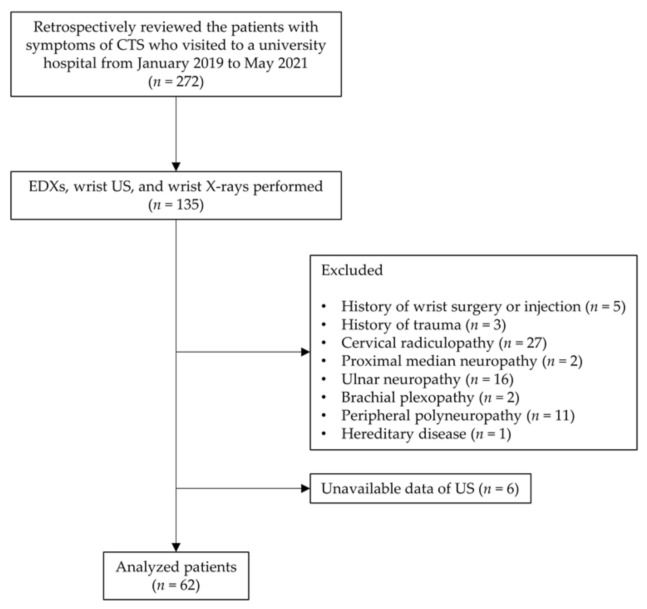
Study flow chart. Abbreviations: CTS, carpal tunnel syndrome; EDXs, electrodiagnostic studies; US, ultrasonography.

**Figure 2 jcm-11-02808-f002:**
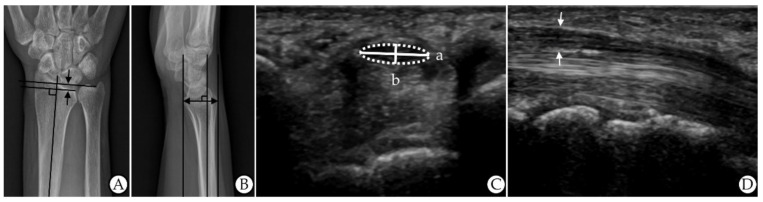
Measurements of ultrasonographic and roentgenographic features. (**A**) Ulnar variance on X-rays (the distance between the two arrows). (**B**) AP diameter of the wrist on X-rays (the distance of the arrow). (**C**) Flattening ratio on US (a (width)/b (height)). (**D**) Thickest AP diameter of the median nerve on US (the distance between the two arrows). Abbreviations: AP, anteroposterior; US, ultrasonography.

**Figure 3 jcm-11-02808-f003:**
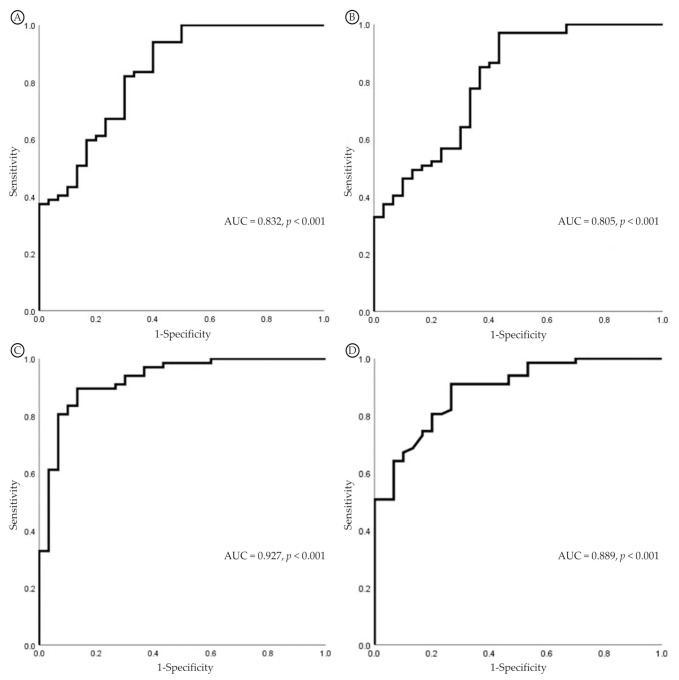
Receiver operating characteristic (ROC) curves for the diagnosis of CTS. (**A**) UV. (**B**) APDW. (**C**) FR. (**D**) TAPDM. Abbreviations: CTS, carpal tunnel syndrome; UV, ulnar variance; APDW, AP diameter of the wrist; FR, flattening ratio; TAPDM, thickest AP diameter of the median nerve; AUC, area under the curve.

**Table 1 jcm-11-02808-t001:** Reclassification of severity grade by electrodiagnostic study.

Severity Grade *	Severity Group
0	Control
1	Mild
2	
3	Moderate
4	Severe
5	
6	

* Severity grades of 0–6 according to the Bland scale.

**Table 2 jcm-11-02808-t002:** Demographic characteristics of the subjects.

		Control	Mild CTS	Moderate CTS	Severe CTS	*p*-Value
Number of Hands		30	34	15	18	
Age (years)		41.13 ± 14.46	53.50 ± 9.05	59.80 ± 8.43	62.89 ± 8.43	<0.001
Sex	Male	11	2	4	6	0.021
	Female	19	32	11	12	
Side	Right	14	20	8	10	0.806
	Left	16	14	7	8	

Values are presented as the mean ± standard deviation. Abbreviation: CTS, carpal tunnel syndrome.

**Table 3 jcm-11-02808-t003:** Pearson’s correlation coefficients between electrodiagnostic parameters and imaging features.

EDXs		X-ray		Ultrasonography	
		UV	APDW	FR	TAPDM
Sensory onset latency		0.358 **	0.333 **	0.772 ***	0.458 ***
Sensory peak latency		0.334 **	0.332 **	0.772 ***	0.438 ***
Sensory amplitude		−0.443 ***	−0.428 ***	−0.652 ***	−0.370 **
Sensory conduction velocity		−0.361 **	−0.357 **	−0.725 ***	−0.488 ***
Distal motor latency		0.139	0.306 **	0.703 ***	0.434 ***

** *p* < 0.01, *** *p* < 0.001. Abbreviations: EDXs, electrodiagnostic studies; UV, ulnar variance; APDW, AP diameter of the wrist; FR, flattening ratio; TAPDM, thickest AP diameter of the median nerve.

**Table 4 jcm-11-02808-t004:** Pearson’s correlation coefficient between ultrasonographic and roentgenographic features.

		X-ray		Ultrasonography	
		UV	APDW	FR	TAPDM
X-ray	UV	1			
	APDW	0.310 **	1		
Ultrasonography	FR	0.320 **	0.467 ***	1	
	TAPDM	0.283 *	0.391 ***	0.320 **	1

* *p* < 0.05, ** *p* < 0.01, *** *p* < 0.001. Abbreviations: UV, ulnar variance; APDW, AP diameter of the wrist; FR, flattening ratio; TAPDM, thickest AP diameter of the median nerve.

**Table 5 jcm-11-02808-t005:** Roentgenographic and ultrasonographic features in CTS patient and control groups.

		Control Group	CTS Patient Group	*p*-Value
UV		−0.01 ± 1.55	1.45 ± 1.89	<0.001
APDW (mm)		22.14 ± 2.07	23.81 ± 2.34	0.001
FR		2.81 ± 0.28	3.53 ± 0.52	<0.001
TAPDM (mm)		1.97 ± 0.35	2.44 ± 0.46	<0.001

Values are presented as the mean ± standard deviation. Abbreviations: UV, ulnar variance; APDW, AP diameter of the wrist; FR, flattening ratio; TAPDM, thickest AP diameter of the median nerve.

**Table 6 jcm-11-02808-t006:** Roentgenographic and ultrasonographic features in four severity groups.

		Control Group	Mild Group	Moderate Group	Severe Group	*p*-Value
UV		−0.01 ± 1.55 ^c^	1.69 ± 2.18	0.96 ± 1.61	1.41 ± 1.61 ^c^	0.003
APDW (mm)		22.14 ± 2.07 ^b^	23.11 ± 2.29	23.85 ± 1.93 ^b^	25.09 ± 1.93	<0.001
FR		2.81 ± 0.28 ^a, b, c^	3.24 ± 0.41 ^a, d, e^	3.80 ± 0.48 ^b, d^	3.87 ± 0.48 ^c, e^	<0.001
TAPDM (mm)		1.97 ± 0.35 ^a, b, c^	2.34 ± 0.39 ^a^	2.49 ± 0.52 ^b^	2.58 ± 0.52 ^c^	<0.001

Values are presented as the mean ± standard deviation. In post-hoc analysis, ^a^
*p* < 0.05 in control group vs. mild group, ^b^
*p* < 0.05 in control group vs. moderate group, ^c^
*p* < 0.05 in control group vs. severe group, ^d^
*p* < 0.05 in mild group vs. moderate group, and ^e^
*p* < 0.05 in mild group vs. severe group. Abbreviations: UV, ulnar variance; APDW, AP diameter of the wrist; FR, flattening ratio; TAPDM, thickest AP diameter of the median nerve.

**Table 7 jcm-11-02808-t007:** Univariate logistic regression analysis and multiple logistic regression analysis using stepwise backward elimination.

	Univariate Logistic Regression				Multiple Logistic Regression			
	Odds Ratio	95% Confidence Interval		*p*-Value	Odds Ratio	95% Confidence Interval		*p*-Value
		Lower	Upper			Lower	Upper	
UV	1.43	1	2.04	0.05				
APDW	1.16	0.91	1.48	0.237				
FR	86.52	9.26	808.83	<0.001	52.52	5.5	501.73	0.001
TAPDM	15.33	2.78	84.62	0.002	8.91	1.2	65.97	0.032

Abbreviations: UV, ulnar variance; APDW, AP diameter of the wrist; FR, flattening ratio; TAPDM, thickest AP diameter of the median nerve.

## Data Availability

The datasets used and/or analyzed in this study are available from the corresponding author upon reasonable request.

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
