# Peer review of "Correlation between Electrodiagnostic Study and Imaging Features in Patients with Suspected Carpal Tunnel Syndrome"

_jcm, 2022, doi:10.3390/jcm11102808_

Round 1

Reviewer 1 Report

The authors studied 62 patients (97 wrists) with suspected carpal tunnel syndrome (CTS). Patients underwent nerve conduction studies, ultrasound of the wrist, and wrist X-rays as well. The parameters derived from X-rays were ulnar variance (UV) and the anteroposterior diameter of the wrist (APDW). All electrodiagnostic parameters were statistically correlated with the roentgenographic and US findings. The authors conclude, that both wrist X-rays and wrist ultrasound could be useful for the diagnosis of CTS as supplements to electrodiagnostic studies.

This is a quite interesting study. That these roentgenographic measurements are correlated to the nerve conduction studies is new for me.

Plausible is for me that there may be a kind of predisposition for a narrow carpal canal, which could be estimated by X-rays. Or: are there some predisposing diseases known which may change these X-ray parameters such as rheumatic diseases?

I think that the study has been carefully done.

Nevertheless, the authors conclude that wrist X-rays may be a potentially useful tool in the diagnosis of CTS. I have some doubts about that. Gold standard for diagnosis are clinically presentation an electrodiagnostic testing. The question is: Are wrist X-rays suited to make a diagnosis of CTS of its own? ROC curve analysis showed some good results for sonographic parameters as shown in the manuscript. What is regarding the X-ray parameters? It should also be possible to analyse ROC curves for X-ray parameters.

In addition, I miss R² and corrected R² values in the regression models.

Reviewer 2 Report

very interesting study on how to correlate the x-ray images to the emg in the diagnosis of STC.
In my opinion I do not give the patient extra radiation if the diagnosis is clear with the emg. On the other hand , the data presented by the authors are nevertheless innovative and noteworthy.

Reviewer 3 Report

This paper compared ultrasonography (US) and roentgenographic results with electrodiagnostic (EDX) data in patients with carpal tunnel syndrome (CTS) and showed there are statistical correlation between imaging and EDX parameters, suggesting the potentials of ultrasonography and X-rays as supplementary diagnostic tool for CTS. This study performed solid statistical analysis of data from 68 patient, including ANOVA test for groups with different severity, t-test between CTS patient and control groups, Pearson’s correlation analysis between EDX and imaging parameters.

Several comments may be addressed before being considered for publication.

  • The authors may consider revising the title of this paper since ‘relationship’ does not highlight the meaningful correlation between EDX and imaging results.
  • The authors used ‘findings’ throughout the manuscript. May consider replacing it to ‘features’ or ‘attributes’?
  • In Abstract line 25-26, please revise the sentence since it is a little bit confusing.
  • Please add more details in the caption of Figure 2 since the A and B are X-ray images, C and D are US images. For example, in A, is ulnar variance the distance between the two black arrows? In C, assuming a is the width and b is the height of the ellipse?
  • In 3.2, can the authors explain more what a negative correlation means? And what does a higher correlation mean since it varies from 0.3 to 0.7?
  • In table 3, it looks like US has stronger correlation with EDXs than X-ray. Does that mean US is better than X-ray diagnostic tool for CTS? Can the authors discuss it?
